# Achievements and challenges of lymphatic filariasis elimination in Sierra Leone

**Yakuba M. Bah[1], Jusufu Paye[2], Mohamed S. Bah[2], Abdulai Conteh[1], Victoria Redwood-Sawyerr[2], Mustapha Sonnie[2], Amy Veinoglou[3], Joseph B. Koroma[4], Mary H. Hodges[2]\*, Yaobi Zhang[5]**

**1** Neglected Tropical Disease Program, Ministry of Health and Sanitation, Freetown, Sierra Leone, **2** Helen Keller International, Freetown, Sierra Leone, **3** Helen Keller International, New York, United States of America, **4** Family Health International 360, Accra, Ghana, **5** Helen Keller International, Regional Office for Africa, Dakar, Senegal

\* mhodges@hki.org

**Data Availability Statement:** Data are stored in the NTD database at the Ministry of Health and Sanitation, Sierra Leone and are available with certain restrictions due to the patients personal

## Abstract

### Background

Lymphatic filariasis (LF) is targeted for elimination in Sierra Leone. Epidemiological coverage of mass drug administration (MDA) with ivermectin and albendazole had been reported >65% in all 12 districts annually. Eight districts qualified to implement transmission assessment survey (TAS) in 2013 but were deferred until 2017 due to the Ebola outbreak (2014–2016). In 2017, four districts qualified for conducting a repeat pre-TAS after completing three more rounds of MDA and the final two districts were also eligible to implement a pre-TAS.

### Methodology/Principal findings

For TAS, eight districts were surveyed as four evaluation units (EU). A school-based survey was conducted in children aged 6–7 years from 30 clusters per EU. For pre-TAS, one sentinel and one spot check site per district (with 2 spot check sites in Bombali) were selected and 300–350 persons aged 5 years and above were selected. For both surveys, finger prick blood samples were tested using the Filariasis Test Strips (FTS).

For TAS, 7,143 children aged 6–7 years were surveyed across four EUs, and positives were found in three EUs, all below the critical cut-off value for each EU. For the repeat pre-TAS/pre-TAS, 3,994 persons over five years of age were surveyed. The Western Area Urban had FTS prevalence of 0.7% in two sites and qualified for TAS, while other five districts had sites with antigenemia prevalence >2%: 9.1–25.9% in Bombali, 7.5–19.4% in Koinadugu, 6.1–2.9% in Kailahun, 1.3–2.3% in Kenema and 1.7% - 3.7% in Western Area Rural.

### Conclusions/Significance

Eight districts in Sierra Leone have successfully passed TAS1 and stopped MDA, with one more district qualified for conducting TAS1, a significant progress towards LF elimination.

information contained in the data. Those who are interested should contact Dr. I. Kargbo-Labour, NTDP Manager, Ministry of Health and Sanitation, Sierra Leone "kargbolabour@gmail.com".

**Funding:** These surveys were made possible with funding from the United States Agency for International Development (USAID) through a grant to Helen Keller International, Cooperative Agreement No. GHS-A-00-06-00006-00 with the End NTDs in Africa project managed by Family Health International 360. The contents are the responsibility of the authors and do not necessarily reflect the views of USAID or the United States Government. The funders had no role in the study design, data collection and analysis, decision to publish, or preparation of the manuscript.

**Competing interests:** The authors have declared that no competing interests exist.

However, great challenges exist in eliminating LF from the whole country with repeated failure of pre-TAS in border districts. Effort needs to be intensified to achieve LF elimination.

## Author summary

Lymphatic filariasis or elephantiasis is targeted for elimination in Sierra Leone, with annual mass treatment with ivermectin and albendazole, and required coverage was achieved in all 12 districts annually. In 2017, transmission assessment survey (TAS) was conducted in eight districts to assess whether treatment can be stopped and pre-TAS was conducted in six other districts to assess whether TAS can be conducted. Eight TAS districts were surveyed as four evaluation units (EU), and a school-based survey was conducted in 1703–1926 children aged 6–7 years from 30 clusters per EU. Six pre-TAS districts were surveyed with one sentinel and one/two spot check sites per district and 300–350 persons aged ≥5 years were tested. All tests were using the Filariasis Test Strips with finger prick blood samples. There were 0–7 positive cases in each TAS EU respectively, all below the critical cut-off value, confirming that mass treatment was no longer needed in these eight districts, a significant progress towards LF elimination. One district had prevalence of <1% in two sites and qualified for TAS, while other five districts had sites with prevalence >2%, suggesting that mass treatment needs to continue. Repeated failure of pre-TAS poses great challenge to eliminate LF in Sierra Leone.YMB is the former NTDP manager and coordinated the MDAs with AC and MS. JBK designed and oversaw the early NTDP and baseline surveys. JP, MB designed and led the TAS, pre-TAS field work data collection. MB and VRS conducted the data analysis. VRS and MH reanalysed the coverage data. YZ produced the point prevalence map. MH drafted the manuscript. MH and YZ revised the manuscript. All authors reviewed and approved the final manuscript.

## Introduction

Lymphatic filariasis (LF) or elephantiasis is a mosquito-borne disease caused by infection with filarial parasites *Wuchereria bancrofti*, *Brugia malayi* or *Brugia timori*. The disease affects poor and marginalised communities leading to disability and disfigurement manifested as hydrocoele, adeno-lymphangitis and lymphoedema [1]. LF leads to diminished productivity, reduced life expectancy and perpetuation of poverty. In 2000, the Global Programme to Eliminate LF was launched and a Global Alliance was established to help endemic countries interrupt transmission and alleviate/prevent related disability [2]. The main strategy to interrupt LF transmission has been annual mass drug administration (MDA) to eligible persons in endemic areas. Exclusion criteria include children under 5 years of age (less than 90cm tall), pregnant women and those that are less than two-weeks post-partum, the very old and the very frail. To be effective, MDA must achieve 100% geographic coverage of endemic areas and ≥65% epidemiological coverage of at-risk populations or ≥80% programmatic coverage of targeted eligible populations (which excludes those not eligible for MDA). World Health Organization (WHO) recommends that 5–6 rounds of effective annual MDA be conducted to eliminate LF transmission [3]. However, in practice this has been insufficient in many endemic countries where pockets of hotspots persist even after 10 or more rounds [4].

It was estimated that LF infected 120 million people in 73 countries with 44 million persons having clinical manifestations [5]. By 2018, 24 LF endemic countries (of 73) no longer required

MDA and were conducting post-MDA surveillance and global estimation of people requiring LF MDA dropped from 1.4 billion in 2011 to 893 million in 2018 [6].

In the 1990s the endemicity of LF in Sierra Leone was one of the highest in Africa [7] with Moyamba district having the highest microfilaremia (mf) prevalence (34.8%) [8]. In 2005, mapping conducted in Sierra Leone with Immunochromatographic test (ICT) cards showed that all 14 districts were LF endemic, among which 12 were co-endemic with onchocerciasis (except Western Area Rural and Western Area Urban) [9,10]. In 2006 community-directed treatment with ivermectin (CDTI) for onchocerciasis commenced in meso- and hyper-endemic villages using volunteer community drug distributors (CDDs) and in 2007 CDTI plus albendazole was piloted in six districts then expanded to all 12 co-endemic districts in 2008 [11].

In rural areas, MDA was implemented by unpaid CDDs selected by their communities with an average 250 persons to be targeted per CDD. The CDDs are trained and supervised by peripheral health workers to update their village census, perform MDA to all eligible individuals and report any adverse events. The CDDs administered between 1 and 4 ivermectin tablets (4 mg) depending on the height of the person by using a dose pole while only one tablet of albendazole (400 mg) was co-administered to each eligible person. The LF MDA lasted for six-eight weeks wherein CDDs treat community members in the morning or late in the evening when they are more easily accessible before or after work. Communities that have schools are targeted during school hours with support from teachers to help organize the pupils.

In 2009, the CDTI plus albendazole approach using CDDs in the remaining 2 districts (Western Area Urban and Western Area Rural) was found to be ineffective in these urban/rapidly urbanizing settings attaining only 29% epidemiological coverage. So, in 2010 a distribution strategy based upon a five days-immunisation campaign was introduced implemented by community health workers (CHWs)/trainee health workers paid a minimal fee commensurate with other programs such as vitamin A supplementation, polio immunisation, Ebola contact-tracing/sensitization [12]. Since 2011, CHWs/trainees have also been paid to conduct MDA all 12 districts headquarter towns.

Since 2011, independent monitoring has been performed during MDA to identify poor performance and get corrective action established and at the end of MDA to validate reported coverage [13,14]. Where program coverage was found to be ineffective the results were shared immediately with the local Peripheral Health Unit (PHU) staff, the District Health Management Teams (DHMTs) and National NTD Program (NTDP) for corrective action and 'mopping-up'.

In 2013, after five rounds of effective MDA, a pre-transmission assessment survey (pre-TAS) using microscopy for *W. bancrofti* mf from 'midnight' blood samples were conducted in the 12 rural districts (except Western Area). The results showed that eight districts (Bo & Pujehun, Bonthe & Moyamba, Kono & Tonkolili, Port Loko & Kambia) had mf prevalence less than 1% at each survey site and qualified to implement transmission assessment surveys (TAS) [15]. Four districts (Bombali & Koinadugu, Kailahun & Kenema) had mf prevalence greater than 1% in at least one site and needed to implement further two rounds of effective MDA and repeat the pre-TAS [15].

Due to the Ebola outbreak (2014–2016), the TAS surveys in the 8 qualified districts were deferred to 2017, after another three rounds of LF MDA due to perceived risks of Ebola transmission and probable lack of community compliance. The four districts that did not qualify for TAS in 2013 also received additional three rounds of LF MDA and underwent repeat pre-TAS in 2017, and the last two districts (Western Area Rural and Western Area Urban) had performed 6 rounds of LF MDA and implemented their first pre-TAS in the same year.

In total, 6–8 rounds of MDA were implemented in these districts respectively. According to the WHO recommendations, these districts should have reached the critical objectives of

stopping LF transmission hence no longer requiring MDA or qualifying for conducting TAS. The purpose of the surveys was to test whether the respective objectives had been reached in these districts. This paper presents the results of the surveys in the context of their MDA coverage data and discusses the challenges and recommendations for LF elimination in Sierra Leone.

## Methods

### Ethics statement

The TAS and pre-TAS were part of the monitoring and evaluation activities of the national LF elimination programme and was conducted by the NTDP of the Ministry of Health and Sanitation (MOHS) Sierra Leone per WHO recommendations. Ethical approval for the survey was obtained from the MOHS Research and Ethics Committee. Prior to the surveys, letters from the national NTDP to the traditional authorities requesting permission to carry out the survey were delivered by the DHMTs. Oral informed consent was first obtained from head teachers and/or village chiefs and then from parents/guardians on behalf of each child and by each child participant before samples were collected and their acceptance was recorded on a form by the leader of the survey team, as literacy rates are low in the country. All participants were eligible for inclusion without discrimination on gender, social status, religion or ethnicity. Recruitment was voluntary, and participants were encouraged to ask questions concerning the survey and their right not to participate and/or to withdraw at any time without repercussions. The data were securely stored in the NTDP database and no identity of participants can be revealed upon publication of this paper.

For TAS the district education officers organized a meeting to sensitise the head teachers of the selected primary schools. At community level, with the help of peripheral health unit staff, community meetings were held in each school to sensitize stakeholders including village headmen, youth groups, village health committee members, parents and teachers.

### MDA coverage data collection

During MDA, CDDs distributed drugs and recorded persons treated. The number of persons treated by CDDs and the total number of persons in communities were summarized and reported to the PHUs. These numbers within the PHUs were then summarized and reported to districts, and in turn to the NTDP. Such reported national MDA results were stored in the NTDP database. Independent monitors also conducted coverage evaluation at the end of MDA. Independent monitoring was based upon the polio monitoring tool of 30 sites and 30 interviews per unit [13,14]. Sites were purposefully selected in hard to reach communities to ensure the most vulnerable are assessed.

### Transmission assessment survey (TAS)

TAS was conducted in the 4 evaluation units (EUs, 8 districts) that passed pre-TAS in 2013. The Microsoft Excel-based Survey Sample Builder (SSB) tool was used to determine sample sizes and selection following the WHO guidelines [3]. As the reported primary school enrolment rate was above 75% in all 4 EUs a school-based cluster survey was conducted. A comprehensive list of primary schools was provided by the Ministry of Education, Science and Technology. Schools were numbered sequentially according to geographical proximity. A list of random cluster numbers was generated by the SSB and the schools with the corresponding numbers were selected as primary or supplementary survey schools. The SSB was also used to calculate appropriate sampling interval and random starting number to generate two lists: A and B. The sample size of 1,692 children aged 6–7 years were targeted in each EU.

Teams worked with school authorities to assemble eligible children in classes 1 and 2 (aged 6–7 years) by sex and sensitized the children about the survey. List A or List B was randomly selected and children in lines of male and female were selected according to the numbers on the list until the next number on the list was higher than the total number of children. Demographic data (name, age, sex) were collected from each selected child who were immediately informed about their test results in the presence of their teachers.

Three days trainings (two laboratory days and one field day) were conducted for all technicians, team members and supervisors, to ensure that protocols and procedures were understood. Each survey team was comprised of a supervisor, a team leader, two technicians and a support staff, and surveyed one district (2 teams per EU). In addition, one CHW was recruited locally per site to assist with communication in the local language and sensitisation with community leaders.

## Pre-transmission assessment survey (pre-TAS)

The repeat pre-TAS was conducted in the four districts that failed pre-TAS in 2013 and a pre-TAS was conducted in both districts in the Western Area. In each district one previously surveyed sentinel site and one spot check site was selected. Since Bombali district had the highest mf prevalence at baseline and pre-TAS, one additional spot check site was selected. Spot check sites were purposefully selected in villages known to have many lymphedema and hydrocele cases. A convenience sample of a minimum of 300 participants over five years of age were recruited at each site following WHO guidelines. If the sample size could not be reached at the primary village, the team moved to a neighbouring village until the sample size was reached.

## Microfilaria detection

The FTS positive cases from the repeat pre-TAS survey in the four districts were followed up for microfilaria detection. Both midday and midnight blood samples were collected. Thick blood films were prepared [8] and examined for microfilariae (mf) of *W. bancrofti* or *Mansonella perstans*. Briefly, fingertip blood samples (60μl) were collected, smeared gently and uniformly onto a slide in a circular shape and allowed to air dry at room temperature for 12–24 hours. The dried smear was dehaemoglobinized next day through flooding with distilled water for 3–5 minutes, air-dried again, and fixed with methanol for 30–60 seconds. These were then stained with GIEMSA for 10 minutes and examined for mf under a light microscope by experienced technicians. For quality control, all slides were double checked by a senior technician from Ghana.

## Antigenemia diagnostic tests

For both surveys, the Alere Filariasis Test Strips (FTS) were used to detect *W. bancrofti* antigen following the manufacturer's instructions. The technicians collected 75μl of blood by fingerprick using sterile lancets and micropipettes. The result of each test was read 10 minutes after adding the blood samples. All positive results from the first tests were confirmed by a second FTS test as recommended by WHO. A person was considered positive only when both first and second tests were positive.

## Data analysis

Epidemiological coverage of the reported MDA results was calculated first using the CDD census data in rural areas adjusted according to recent vaccination campaign data in urban settings as denominators [16] and second using the projected population data from the 2015

population census as denominators [17]. Program coverage was obtained from the independent monitoring coverage evaluation. The calculated coverage was evaluated as effective or ineffective against the WHO recommended thresholds of 65% for epidemiological coverage and 80% for program coverage.

The geographical location of each survey site was plotted using the global positioning system coordinates collected for each site. The prevalence of antigenemia was calculated by dividing the number of FTS positives by the total viable samples tested per site. An EU was deemed to have passed the TAS if the number of FTS positives was below the critical cut-off value of 20 calculated by the SSB. The district was deemed to have passed the pre-TAS if the prevalence of FTS positives was <2% at each site or to have failed the pre-TAS if it was ≥2% in any site. ArcGIS version 10.6 (ESRI, Redlands, California, United States) was used to plot the maps and analysis on the clustering effect of the FTS positive cases was conducted using the Spatial Autocorrelation (Moran's Index) in the package.

## Results

### MDA coverage results by district

All 12 districts have reported effective epidemiological coverage (≥65%) using the national census projection as denominator in every round as shown in Fig 1. When using the CDD-generated census data as denominator, all districts reached effective epidemiological coverage (≥65%) in each of three rounds of MDA, except two TAS districts where Bonthe had ineffective epidemiological coverage (<65%) in 2 of 3 years and Pujehun had ineffective epidemiological coverage (<65%) in all 3 years. According to the independent monitoring of program coverage in three years, among the four pre-TAS districts (left panel, Fig 1), Bombali and Kailahun achieved effective program coverage (≥80%) in all three years, whilst Koinadugu and Kenema achieved this in only 1 of 3 years. Among the eight pre-TAS districts (right panel, Fig 1), only Moyamba achieved effective program coverage (≥80%) in three years; three districts (Kambia, Kono and Tonkolili) achieved this target in two years; one district (Bo) achieved this target in only one year; and three districts (Pujehun, Bonthe and Port Loko) did not achieve effective program coverage (≥80%) in any year.

### TAS in eight districts

Overall, 7,1057,105 children aged 6–7 years were tested (males: 49.5%, females: 50.5%) in four EUs. A total of 9 FTS positives (males: 4, females: 5) were found: 7 in Kono/Tonkolili, 1 in Port Loko/Kambia and 1 in Bo/Pujehun, while no positive was found in the other EU as shown in Table 1. The number of FTS positives was well below the critical cut-off value for each EU, successfully passing the TAS. The locations of the surveyed clusters (schools) and the positive cases were shown in Fig 2. There was no clustering effect of the positive cases (Moran's I = -0.04, Z-score = -0.02, $p$ = 0.84).

### Repeat pre-TAS in four districts and pre-TAS in two districts

Overall, 3,994 persons over five years of age were tested (males: 36.8%% and females: 63.2%%), 2,744 in the four districts during repeat pre-TAS and 1,250 in 2 pre-TAS districts as shown in Table 2. The individual site antigenemia prevalence was 9.1%, 16.7% and 25.9% in Bombali, 7.5% and 19.4% in Koinadugu, 6.1% and 2.9% in Kailahun, and 1.3% and 2.3% in Kenema. There was no significant difference in positivity by sex. All four repeat pre-TAS districts failed again to reach the threshold of <2% antigenemia prevalence for qualifying for TAS. For the pre-TAS the individual site antigenemia prevalence was 0.7% and 0.7% in Western Area

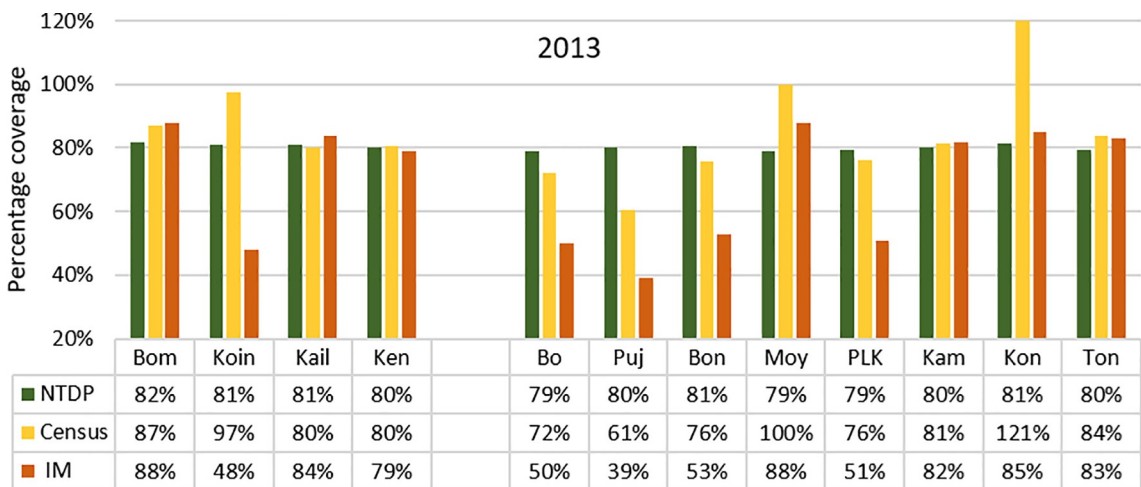

| | Bom | Koin | Kail | Ken | | Bo | Puj | Bon | Moy | PLK | Kam | Kon | Ton |
|---|---|---|---|---|---|---|---|---|---|---|---|---|---|
| NTDP | 82% | 81% | 81% | 80% | | 79% | 80% | 81% | 79% | 79% | 80% | 81% | 80% |
| Census | 87% | 97% | 80% | 80% | | 72% | 61% | 76% | 100% | 76% | 81% | 121% | 84% |
| IM | 88% | 48% | 84% | 79% | | 50% | 39% | 53% | 88% | 51% | 82% | 85% | 83% |

**Districts**

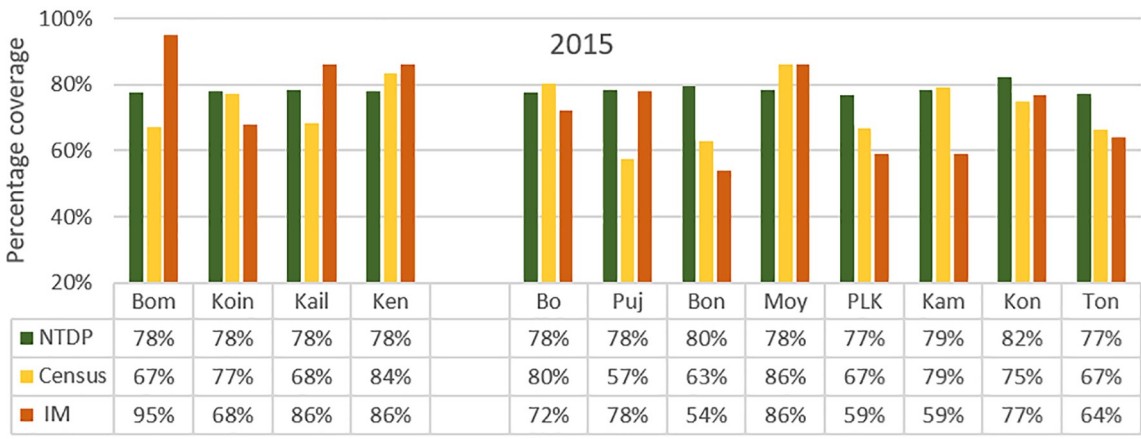

| | Bom | Koin | Kail | Ken | | Bo | Puj | Bon | Moy | PLK | Kam | Kon | Ton |
|---|---|---|---|---|---|---|---|---|---|---|---|---|---|
| NTDP | 78% | 78% | 78% | 78% | | 78% | 78% | 80% | 78% | 77% | 79% | 82% | 77% |
| Census | 67% | 77% | 68% | 84% | | 80% | 57% | 63% | 86% | 67% | 79% | 75% | 67% |
| IM | 95% | 68% | 86% | 86% | | 72% | 78% | 54% | 86% | 59% | 59% | 77% | 64% |

**Districts**

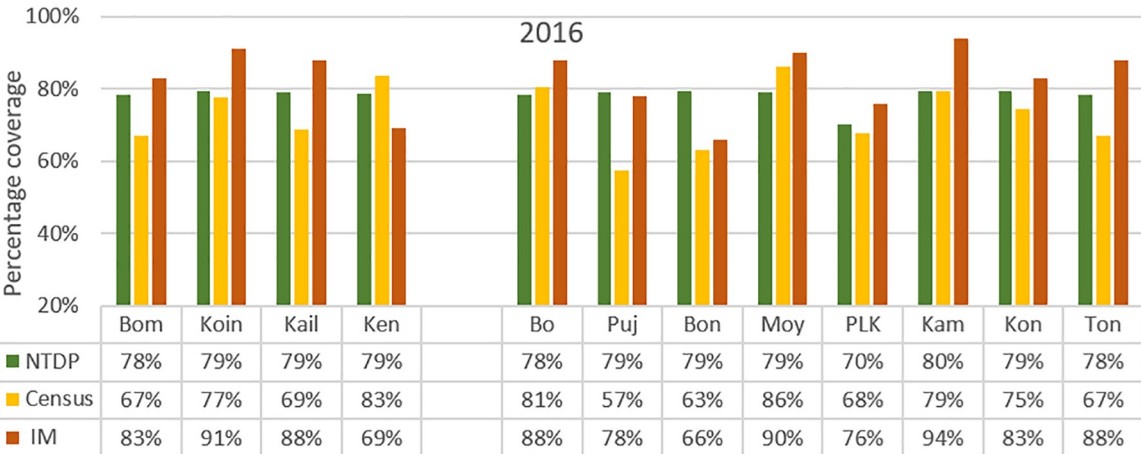

| | Bom | Koin | Kail | Ken | | Bo | Puj | Bon | Moy | PLK | Kam | Kon | Ton |
|---|---|---|---|---|---|---|---|---|---|---|---|---|---|
| NTDP | 78% | 79% | 79% | 79% | | 78% | 79% | 79% | 79% | 70% | 80% | 79% | 78% |
| Census | 67% | 77% | 69% | 83% | | 81% | 57% | 63% | 86% | 68% | 79% | 75% | 67% |
| IM | 83% | 91% | 88% | 69% | | 88% | 78% | 66% | 90% | 76% | 94% | 83% | 88% |

**Districts**

**Fig 1. District-level treatment coverage during 2013–2016 in Sierra Leone NTDP: epidemiological coverage of the national reported treatment among the district total population projected from the 2015 national census; Census: epidemiological coverage of the national reported treatment among the district total population from the CDD census; IM: program coverage from independent monitoring evaluation.** Left panel: four repeat pre-TAS districts Bombali (Bom), Koinadugu (Koin), Kailahun (Kail) and Kenema (Ken); Right panel: eight TAS districts Bo, Pujehun (Puj), Bonthe (Bon), Moyamba (Moy), Port Loko (PLK), Kambia (Kam), Kono (Kon) and Tonkoulili (Ton).

Urban qualifying for conducting TAS and 1.7% and 3.7% in Western Areas Rural failing to qualify for TAS as shown in Fig 3 and Table 2.

## Microfilaria detection in the FTS positive cases

In order to investigate potential interference of *M. perstans* infections on FTS tests, a follow-up study was conducted before the next MDA by tracing the 277 FTS positive cases in the four repeat pre-TAS districts. Among these, 236 provided a midnight blood sample and 232 provided a midday blood sample (80% response rate). Nine persons were positive for *W. bancrofti* mf (4 in Bombali and 5 in Koinadugu) in the 236 midnight blood samples and 12 persons were positive for *M. perstans* mf (2 in Bombali, 9 in Koinadugu and 1 in Kailahun) in the 232 midday blood samples. Six samples were positive for both *W. bancrofti* and *M. perstans* mf, six samples were positive for *M. perstans* mf and negative for *W. bancrofti* mf and three samples were positive for *W. bancrofti* mf and negative for *M. perstans* mf.

## Discussion

Eight of the 14 endemic districts in Sierra Leone successfully passed TAS and reached the criteria of stopping LF MDA and another district has qualified for conducting TAS. A total of 3.8 million people in Sierra Leone now no longer require LF MDA. Despite this achievement, five districts failed to reach the criteria for conducting TAS, in particular, four districts (Bombali, Kailahun, Kenema and Koinadugu) failed pre-TAS twice. This is despite the 6–8 rounds of MDA with good reported coverage. Major challenges exist towards achieving LF elimination in the whole country.

Bombali, Kailahun, Kenema and Koinadugu are border districts with Guinea (Bombali Koinadugu and Kailahun) and Liberia (Kailahun and Kenema). These four districts had relatively high baseline antigenemia prevalence as shown in Table 2. Repeated in-migration of infected individuals for traditional healing in Bombali, Koinadugu and the Western Area Rural attract LF sufferers from the West Africa region for traditional management in secretive locations. These reclusive communities may have created foci of persistent infection. The WHO

**Table 1. Results of transmission assessment surveys (TAS) in eight districts in four EUs in Sierra Leone in 2017.**

| EU | District | Baseline mapping in 2005 | Number of persons tested per EU | | Number of FTS positives | | Critical cut-off value | | | |
|------|-----------|-----------------------------|----------|----------|----------|-------|-------|---------|-------|----|
| | | Prevalence (%) by ICT | Males | Females | | Total | Males | Females | Total | |
| EU 1 | Kambia | 15.5 | 970 | 958 | | 1,926 | | 1 | 1 | 20 |
| | Port Loko | 20.5 | | | | | | | | |
| EU2 | Bo | 15 | 868 | 881 | | 1,749 | | 1 | 1 | 20 |
| | Pujehun | 4.4 | | | | | | | | |
| EU3 | Kono | 30 | 889 | 836 | | 1,725 | 4 | 3 | 7 | 20 |
| | Tonkolili | 37 | | | | | | | | |
| EU4 | Moyamba | 10.5 | 792 | 911 | | 1,703 | | | 0 | 20 |
| | Bonthe | 13.1 | | | | | | | | |
| Total | - | - | **3,519** | **3,586** | | **7,105** | **4** | **5** | **9** | |

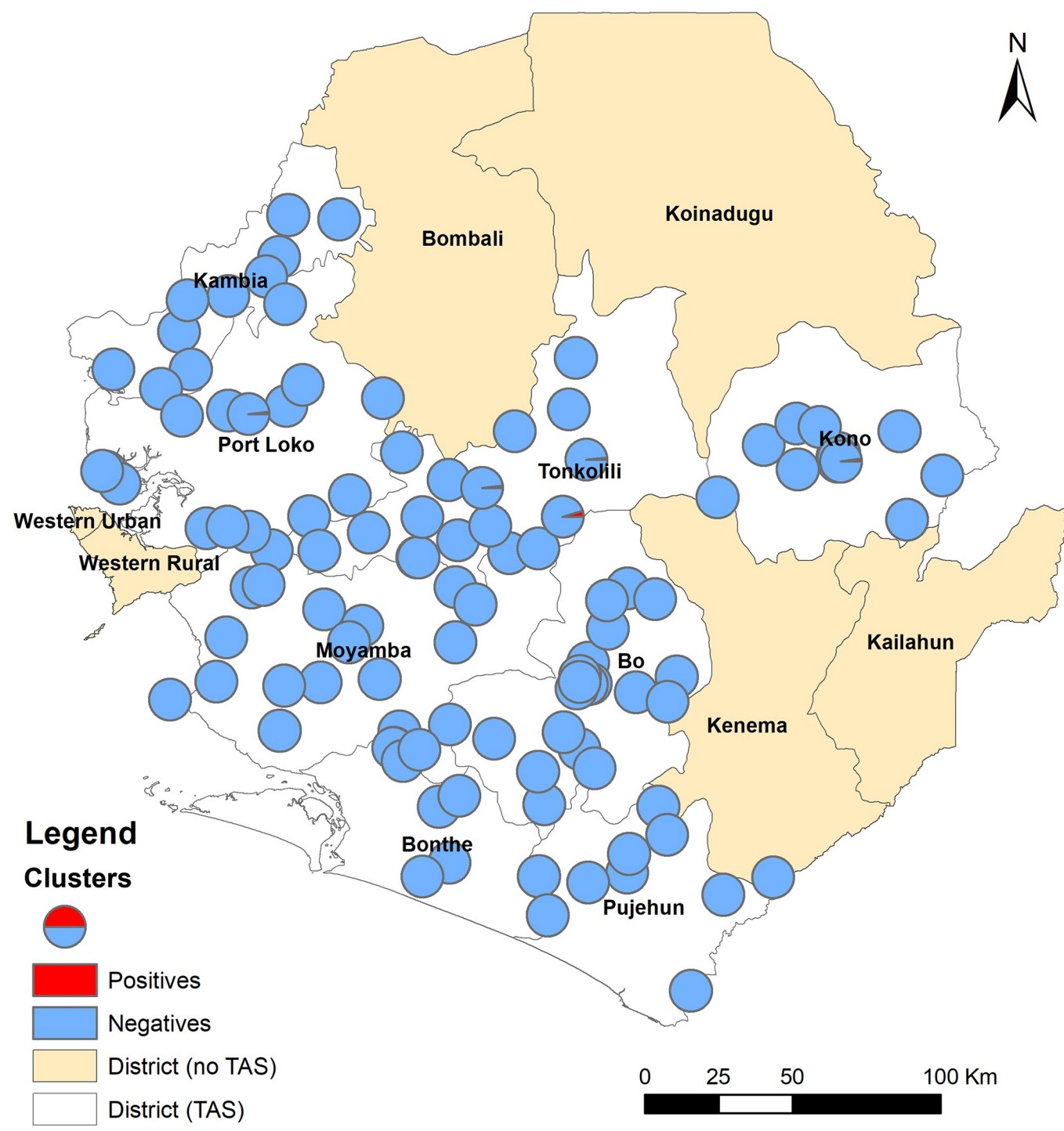

**Fig 2. Distribution of survey clusters and location of FTS positive cases in 8 TAS districts in Sierra Leone in 2017.**

recommends conducting 5–6 rounds of MDA with effective coverage (≥65%) to eliminate LF. While this strategy worked well in many areas with low levels of transmission, it has not worked well in areas with high levels of transmission. Many countries have reported that LF transmission persists in high transmission areas with high baseline prevalence after many

**Table 2. Results of repeat pre-TAS and pre-TAS by survey site in six districts in Sierra Leone in 2017.**

| District | Baseline prevalence (%) by ICT in 2005 * | Type of site | FTS Tested | No of males tested | No of females tested | No of males +ve | No of females +ve | Total positives by FTS | % prevalence by FTS |
|---|---|---|---|---|---|---|---|---|---|
| **Repeat pre-TAS** | | | | | | | | | |
| KAILAHUN | 19.1 (15.0–24.0) | Sentinel | 310 | 147 | 163 | 6 | 13 | 19 | 6.1 |
| | | Spot Check | 310 | 91 | 219 | 7 | 2 | 9 | 2.9 |
| KENEMA | 13.3 (11.7–15.0) | Sentinel | 301 | 174 | 127 | 3 | 1 | 4 | 1.3 |
| | | Spot Check | 300 | 109 | 191 | 3 | 4 | 7 | 2.3 |
| KOINADUGU | 46 (36.0–60.0) | Sentinel | 306 | 112 | 194 | 10 | 13 | 23 | 7.5 |
| | | Spot Check | 304 | 166 | 138 | 21 | 37 | 58 | 19.1 |
| BOMBALI | 52 (38.0–68.0) | Sentinel | 308 | 106 | 202 | 30 | 49 | 79 | 25.6 |
| | | Spot Check | 300 | 116 | 184 | 17 | 33 | 50 | 16.7 |
| | | | 305 | 110 | 195 | 14 | 14 | 28 | 9.2 |
| **Pre-TAS** | | | | | | | | | |
| RWA | 7.2 | Sentinel | 350 | 170 | 180 | 7 | 7 | 14 | 4.0 |
| | | Spot Check | 300 | 41 | 259 | 1 | 4 | 5 | 1.7 |
| UWA | 11.7 | Sentinel | 300 | 77 | 223 | 0 | 2 | 2 | 0.7 |
| | | Spot Check | 300 | 50 | 250 | 0 | 2 | 2 | 0.7 |

Note

* figures in brackets represent minimum and maximum prevalence among sites surveyed in the districts [Ref #9].

years of annual MDA, such as in Nigeria, Tanzania and Ghana [18–20]. In Ghana, after up to 14 rounds of MDA, the transmission of LF still persists in districts with relatively high baseline prevalence [21]. Modelling of LF MDA showed that baseline prevalence is one of the most important factors on impact, along with the number of rounds of MDA [22].

Higher baseline prevalence is possibly related to higher vector competence and redisposing socio-economic circumstances or occupations of the population [23]. The principal mosquito vectors of LF in West Africa of the *Anopheles gambiae* complex with *Culex* species playing only a minor if any role. *Culex* has been found to be the predominant species in the Western Area Urban setting and the South: Bo and Pujehun districts [24]. The importance of vector competence and biting rates contributing to LF hotspots despite repeated effective rounds of MDA has been described in Ghana [25]. The challenges experienced in Sierra Leone are similar to other countries approaching LF elimination targets [26].

Given the relatively high baseline prevalence in these districts in Sierra Leone, it may not be surprising that 8 rounds of MDA still have not reached the point of stopping MDA in these districts. Since the survey, further rounds of MDA have been conducted. It is hoped that these further rounds of MDA would be able to reach the program goals in these districts.

High treatment coverage is critical to achieve the LF elimination goal [27]. Sierra Leone has reported satisfactory MDA coverage over the years. However, this may have been confounded by many factors. After the failure of the repeat pre-TAS survey in the four districts, a qualitative follow up investigation was conducted in Bombali and Koinadugu. It was shown that some communities visited had little knowledge of MDA (authors' program observation). Therefore, despite the good reported coverage at district level, treatment coverage at community or sub-district level may not have been satisfactory. Coverage evaluation through

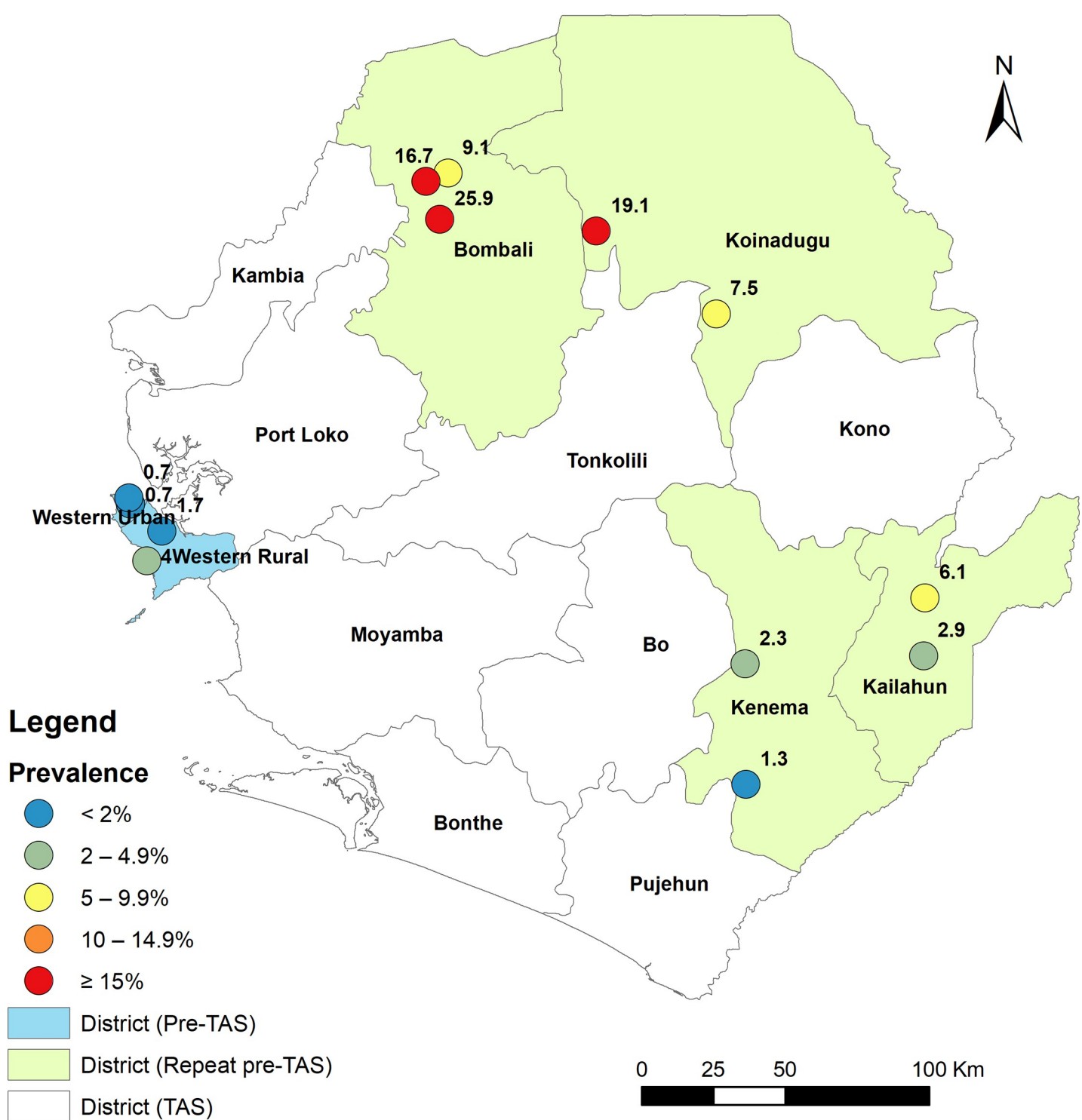

**Fig 3. Location of repeat pre-TAS and pre-TAS survey sites and point-prevalence of FTS positives in 6 districts in Sierra Leone in 2017.**

independent monitoring did show that some districts failed to achieve the required minimum program coverage threshold of 80% as in Fig 1. An in-depth review of sub-district level coverage data would help identify the problematic areas for quality improvement in MDA. Outreach

sensitisation to traditional healers has been implemented and from 2019 traditional healers have also been recruited and trained as community mobilisers and as CDDs to improve the compliance of MDA.

Inaccurate denominators are a well-recognised challenge in Sierra Leone due to rapid un-projected internal employment-seeking migrations between districts, accelerated urbanisation to headquarter towns and the Western Area, persons working away from home, school aged children and young adults attending education facilities away from home and/or from neigh-bouring Liberia [28]. The census data can hardly reflect the reality of the fast-changing population throughout the country. Cattle herders are transitory workers who often only speak a minority language (Fullah). They are less aware of mass campaigns or registered in a village for MDA. The Fullah ethnic group is widespread and functions across the Mano River Union countries and beyond in West Africa. They cross borders frequently as traditional traders and targeting this group of mobile population is a great challenge. It has been promoted to target the cattle herders during 'market days' since 2013 and this should be intensified on both sides of these porous borders perhaps even outside the traditional MDA period and repeatedly in both Bombali and Koinadugu in Sierra Leone and neighbouring districts (Kindia, Mamou and Faranah) in Guinea until MDA is able to be stopped. Synchronisation of LF MDA with neigh-bouring Guinea and Liberia needs to be better coordinated. In future, non-residents will be recorded and reported separately to be able to quantify their impact in the border districts. A buffer drug stock has been sent to these border districts since 2013 to cope with the situation in Sierra Leone and a similar strategy is recommended for Guinea and Liberia.

Inadequate or inconvenient timeframes for MDA made full participation of volunteer CDDs less likely. The best time for CDD participation in MDA was pre-harvest October-December but the best time for MDA is April-May which is preferable for maximum effect on transmission. However, due to approval processes, drug availability in country, competition with other public health priorities and reporting schedules, it is often difficult to keep MDA in the optimal period and in 2019 it was performed at the height of the rainy season.

Reduced CDD motivation in the post-Ebola context where CHWs get paid for similar responsibilities in other programs has become problematic. As of 2019, all CDDs are being paid a commensurate sum for implementation and the timeframe allowed for MDA will be shortened from the previous 6 weeks to a one-week intensive campaign by paid CDDs.

As disease burden and perceived risks fall there has been a reduction in community interest in the LF program and participation by community leaders. Increasingly people believe they are 'not at risk'. For example, as the prevalence of soil transmitted helminths falls and *Ascaris lumbricoides* is now under control, the public believe that the drugs 'don't work' as they do not see roundworms in their stools following albendazole ingestion. In earlier years when disease burden was high, taking albendazole and passing worms was a motivating factor for participation.

Genuine non-compliance/refusals in high baseline districts are often reported as 'fear of side effects' or 'I am not at risk'. Social messaging in the early stages of a NTDP when disease burden is high has been revised regularly for the new context but it has been challenging to explain to the target population that even if they do not consider themselves to be at risk there is a need to participate. Moving social mobilisation from district-level to chiefdom level to enable greater participation of section chiefs and village headmen has been adopted, together with the development and distribution of a pico-video of a testimonial given by a lady with lymphedema encouraging compliance during MDA and explaining how it has helped her condition. Recently a new social messaging strategy has been adopted whereby the NTDP has changed the 'messenger' and are working directly through the organised religious networks ISLAG and CHRISTLAG and traditional healers as they have greater influence over practices

in the most vulnerable communities. In addition, greater supplies of drugs for side effects (itching, swelling, headaches) have been made more widely available.

*M. perstans* is known to co-exist with LF in Sierra Leone [29]. Previous studies in Cameroon and DRC suggested that there is no cross-reactivity of LF antigen with *M. perstans* using ICT cards but with *Loa loa* [30–32]. However, another study also in Cameroon found that in two FTS positive cases, one was positive for *M. perstans* DNA and the other was positive for *Loa loa* mf, while both negative for *W. bancrofti* by mf, Wb123 ELISA and qPCR tests [33]. We therefore tested the FTS positive cases for mf using both midday and midnight blood samples and found that 12 of 232 FTS positives tested were *M. perstans* mf positive and half of these were positive for both *W. bancrofti* and *M. perstans* mf. This suggests that there may be a potential cross-reactivity between *M. perstans* and LF in the FTS tests. Further research is needed on the influence of *M. perstans* on the FTS test and on the way to confirm LF elimination in *M. perstans* co-endemic areas in Sierra Leone.

Koinadugu district has the greatest challenges of poor infrastructure (roads, bridges and communications) and the longest international border (with Guinea). Koinadugu has only 75 PHUs each often with only one staff of the lowest cadre of health worker but serving an average population of 5,458 in catchment area of an average of 162 km$^2$. In recognition of the challenges of Koinadugu (and Bombali) the government has decided to subdivide each of these two districts into two separate districts respectively with 12 new chiefdoms each. This political decision has recently been matched with funds to support separate DHMTs: in Falaba and Karene respectively. Since 2013 additional logistical support has been provided to Koinadugu for implementation and supervision which has been further intensified with national- and district-led teams for combined monitoring sensitisation and MDA in areas most underserved.

An evidence-led vector control program is led by the National Malaria Control Program: four regional hubs provide annual information on vector composition, behavior, susceptibility to insecticides and long-lasting insecticide treated nets (LLITN) longevity and effectiveness. However, despite several rounds of universal distribution the use of LLITN remains low in Sierra Leone. In 2018, the percentage of households with at least one LLITN for every two persons was higher in rural than urban areas (28% versus 21%) and higher in the Eastern and Southern provinces than in other provinces (range 18%-29%) [34]. Public-Private Partnership to scale up the use of insecticide residual spraying as recommended by WHO has not been implemented at scale after a trail period in Bo district in 2012 [35].

One of the two pillars of the global LF elimination strategies is to alleviate the suffering caused by LF through providing a morbidity management package to manage lymphedema and hydrocele [5]. With the progress made in the treatment strategy in reducing prevalence, it has been a great challenge for Sierra Leone to implement the morbidity management strategy, with difficulties in sourcing funding. More effort is needed to build capacity of the country health system to provide services for morbidity management.

There are some limitations of the current study. In the follow up study of the FTS positive cases, some of the positive cases could not be traced and tested, therefore we were not able to estimate the mf prevalence. The sample size was relatively small, and we did not use more sensitive tests, such as WB123 ELISA or qPCR to ascertain the true LF infection in the FTS positive cases.

In conclusion, Eight of the 14 districts in Sierra Leone have successfully passed the TAS and now stopped MDA with one more district qualified to conduct TAS. However, five districts failed the pre-TAS, out of which four districts failed the pre-TAS twice. Sierra Leone is facing challenges in eliminating LF in the remaining districts of the country, such as high baseline prevalence in the border districts, treatment coverage issues due to population movement within the country/across borders, traditional beliefs, demotivated CDDs, reduced motivation

in communities as disease burden and perceived risk fall, vector competence, and potential influence of *M. persta*ns co-endemicity on LF tests, as well as provision of services for morbidity management. Major recommendations include intensified MDA sensitisation and supervision, regionally lead synchronization of MDA with Guinea and Liberia ensuring migratory populations are treated, greater participation of religious leaders and traditional healers, further research to address potential influence of *M. perstans* on LF FTS tests, and effort to source funding for LF morbidity management.

## Supporting information

**S1 Checklist. STROBE Checklist.**
(PDF)

## Acknowledgments

The authors wish to thank WHO for providing technical support to the NTDP and the survey team leaders: A. Tia, S. Saffa; and F. Sahr for the quality control and Angel Weng for graphics. Thanks also go to the district health management teams, head teachers and communities for their collaboration.

## Author Contributions

**Data curation:** Victoria Redwood-Sawyerr, Mary H. Hodges, Yaobi Zhang.

**Formal analysis:** Mohamed S. Bah, Victoria Redwood-Sawyerr, Mary H. Hodges, Yaobi Zhang.

**Investigation:** Jusufu Paye, Mohamed S. Bah.

**Methodology:** Jusufu Paye, Mary H. Hodges, Yaobi Zhang.

**Project administration:** Abdulai Conteh, Mustapha Sonnie, Amy Veinoglou.

**Supervision:** Yakuba M. Bah, Mustapha Sonnie, Joseph B. Koroma.

**Writing – original draft:** Mary H. Hodges.

**Writing – review & editing:** Mary H. Hodges, Yaobi Zhang.

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
