## [Decision Letter · Decision Letter 0]

7 May 2020

Dear Dr. Hodges,

Thank you very much for submitting your manuscript "Achievements and challenges of lymphatic filariasis elimination in Sierra Leone" for consideration at PLOS Neglected Tropical Diseases. As with all papers reviewed by the journal, your manuscript was reviewed by members of the editorial board and by several independent reviewers. In light of the reviews (below this email), we would like to invite the resubmission of a significantly-revised version that takes into account the reviewers' comments. 

Dear Dr Hodges,

three independent experts in filarial MDA elimination programmes have carefully reviewed your manuscript. 

One reviewer has in particular raised concerns regarding the novelty and suitability of the work as a research article for publication in PLoS NTDs.

Because there are many precedents of similar epidemiological surveys pre and post elimination programmes in the PLoS NTDs archive, I do not necessarily agree that this epidemiological survey is unsuitable for publication in PLoS NTDs.

However, the reviewer also queries whether the data set is novel. As it comprises outputs of the LF programme in Sierra Leone including MDA coverage, TAS survey results, and pre-TAS results, the data may have already been put into the public domain elsewhere. The pilot data surveying Mansonella co-prevalence, is not, in my judgement, sufficiently robust for stand alone consideration.

I therefore query with the authors whether the data is unique or has already been published elsewhere? Please consider this carefully in any response to review. 

Could addiitonal analyses be provided to provide further novelty, on clustering effect of TAS positive cases, for instance, as suggested by reviewer 1?

Please also carefully attend to reviewers' critiques on data presentation, especially reviewer 3.

Yours sincerly, Joe Turner, Guest Editor, on behalf of PLoS Editorial team

We cannot make any decision about publication until we have seen the revised manuscript and your response to the reviewers' comments. Your revised manuscript is also likely to be sent to reviewers for further evaluation.

Sincerely,

Joseph D Turner, Ph.D

Guest Editor

Jennifer Keiser

Deputy Editor

Dear Dr Hodges,

three independent experts in filarial MDA elimination programmes have carefully reviewed your manuscript. 

One reviewer has in particular raised concerns regarding the novelty and suitability of the work as a research article for publication in PLoS NTDs.

Because there are many precedents of similar epidemiological surveys pre and post elimination programmes in the PLoS NTDs archive, I do not necessarily agree that this epidemiological survey is unsuitable for publication in PLoS NTDs.

However, the reviewer also queries whether the data set is novel. As it comprises outputs of the LF programme in Sierra Leone including MDA coverage, TAS survey results, and pre-TAS results, the data may have already been put into the public domain elsewhere. The pilot data surveying Mansonella co-prevalence, is not, in my judgement, sufficiently robust for stand alone consideration.

I therefore query with the authors whether the data is unique or has already been published elsewhere? Please consider this carefully in any response to review. 

Could addiitonal analyses be provided to provide further novelty, on clustering effect of TAS positive cases, for instance, as suggested by reviewer 1?

Please also carefully attend to reviewers' critiques on data presentation, especially reviewer 3.

Yours sincerly, Joe Turner, Guest Editor, on behalf of PLoS Editorial team

Reviewer's Responses to Questions

**Key Review Criteria Required for Acceptance?**

**Methods**

-Are the objectives of the study clearly articulated with a clear testable hypothesis stated?

-Is the study design appropriate to address the stated objectives?

-Is the population clearly described and appropriate for the hypothesis being tested?

-Is the sample size sufficient to ensure adequate power to address the hypothesis being tested?

-Were correct statistical analysis used to support conclusions?

-Are there concerns about ethical or regulatory requirements being met?

Reviewer #1: The objectives of the study are not clearly articulated in the methods section, a clear testable hypothesis is not stated. The methods section details the steps the country has taken towards LF elimination as per the GPELF requirements: MDA, coverage assessments, pre-TAS, TAS. The methods section does not provide details of any further statistical analysis, beyond routine reporting of programmatic data.

The MDA epidemiological and programmatic coverage is calculated using three different population sources, it is not clear why these three population sources were selected. Other than presenting the MDA coverage using these three different population sources, there is no analysis presented in the methods to look at differences between the population sources, and the impact this has on achieving the necessary programmatic MDA coverage.

In the ethical approval section of the methods, it states that 'participants identities were protected by collecting, recording and analyzing data such that participants remained anonymous'. However, when detailing where the data can be found, the authors state that 'data is available with certain restrictions due to the patients personal information contained in the data.'

Reviewer #2: (No Response)

Reviewer #3: Please see my comments in the attached word document.

**Results**

-Does the analysis presented match the analysis plan?

-Are the results clearly and completely presented?

-Are the figures (Tables, Images) of sufficient quality for clarity?

Reviewer #1: The results section is very limited, with just over a page of results presented. The results presented are not clear or completely presented. For example, when presenting the MDA coverage results by district - the author refers to districts reporting ineffective coverage 'on a total of 5 of 6 occasions'. 

From table 1, it is evident that districts with a high baseline prevalence (Kono and Tonkolili), have the greatest number of FTS positive cases identified during TAS. However, no analysis is undertaken to determine the impact of MDA on the LF prevalence - the results only present the MDA data (using three population sources), and the TAS/pre-TAS results.

From the map presented in Figure 2 - it is not clear what the cluster represents - is it the location of the household of the positive case or the school? It would be interesting to understand the clustering effect of TAS positive cases - from how many schools did the 7 FTS positive cases identified in Kono and Tonkolili come from? Was information collected during the TAS to determine if these positive individual took the MDA? If so, what were the reasons for not taking MDA? How does the LF programme intend to use these results to plan further surveillance activities.

Reviewer #2: (No Response)

Reviewer #3: Please see my comments in the attached word document.

**Conclusions**

-Are the conclusions supported by the data presented?

-Are the limitations of analysis clearly described?

-Do the authors discuss how these data can be helpful to advance our understanding of the topic under study?

-Is public health relevance addressed?

Reviewer #1: The title of the paper is 'achievements and challenges of LF elimination in Sierra Leone' yet no reference is given to Sierra Leone's achievements and challenges towards the second pillar of the GPELF: morbidity management and disability prevention for lymphodema and hydrocele patients. 

In the discussion, the authors present their 'program observation' to justify potential reasons why the five districts failed pre-TAS, and four districts failed the pre-TAS twice. This is anecdotal evidence and it is not clear how these conclusions have been reached.

Reviewer #2: (No Response)

Reviewer #3: Please see my comments in the attached word document

**Editorial and Data Presentation Modifications?**

Reviewer #1: (No Response)

Reviewer #2: (No Response)

Reviewer #3: I recommend minor revision

**Summary and General Comments**

Reviewer #1: This paper presents the output of the LF programme in Sierra Leone including MDA coverage, TAS survey results, and pre-TAS results. This data presented is not novel and the authors do not conduct any further analysis to explore reasons for pre-TAS failure, other than anecdotal evidence presented in the discussion.

Reviewer #2: This manuscript is well written and reports important outcomes that are highly relevant to the NTD community. The study design and methodology are appropriate for the research question. However, the narrative could be better structured to enhance its impact.

Major comment

The title does not project the value of progressive outcomes that will transition to elimination of LF in Sierra Leone. This manuscript could be the first of three papers in a series titled ‘Towards Elimination of LF in Sierra Leone as suggested below:

I. Towards elimination of LF in Sierra Leone 1: Cross border challenges

II. Towards elimination of LF in Sierra Leone 2: MDA to eliminate LF during public health emergencies

III. Towards elimination of EL in Sierra Leone 3: Interruption of LF transmission nationwide 

The ‘conclusions/significance’ section in the abstract ties in well with the suggested title for the first series: ‘Eight districts in Sierra Leone have successfully passed TAS1 and stopped MDA, with one more district qualified for conducting TAS1, a significant progress towards LF elimination. However, great challenges exist in eliminating LF from the whole country with repeated failure of pre-TAS in border districts. Effort needs to be intensified to achieve LF elimination.

The references to carrying out MDA during the Ebola epidemic should be expanded into a long-awaited story of the contribution of the CDDs in the containment of Ebola in Sierra Leone. This should be a separate paper that explores the following questions: Why were TAS surveys stopped during the Ebola epidemic? How did CDDs contribute to contact tracing and MDA for malaria? What led to the reduced CDD motivation after the Ebola epidemic? It is clear that the data and information to address these questions are available.

Minor comments

Abstract

Line 3: replace ‘every’ with ‘all 12 districts’ to and do the same in the main background. It is a bit confusing, if this clarification is not made from the outset.

Introduction

Paragraph 1, Line 3: replace ‘due to’ with ‘manifested as’.

Paragraph 2, line 1. Why start with 2014? I would limit this to figures for 2017.

Ethical approval

Crosscheck that approval was obtained from MOHS Research and Ethics Committee and not the National Ethics Committee.

Reviewer #3: The topic of the paper is very interesting and important since countries are facing challenges with transmission assessment surveys in some of the settings

PLOS authors have the option to publish the peer review history of their article (what does this mean?). If published, this will include your full peer review and any attached files.

Reviewer #1: No

Reviewer #2: Yes: Moses John Bockarie

Reviewer #3: Yes: Didier Bakajika
---

## [Decision Letter · Decision Letter 1]

13 Oct 2020

Dear Dr. Hodges,

We are pleased to inform you that your manuscript 'Achievements and challenges of lymphatic filariasis elimination in Sierra Leone' has been provisionally accepted for publication in PLOS Neglected Tropical Diseases.

Best regards,

Joseph D Turner, Ph.D

Guest Editor

Jennifer Keiser

Deputy Editor

Dear Authors

apologies for the delay in response to your submitted revised ms. I was awaiting reviewer 1 and 3 comments to your revisions. Having not received such responses, the journal has now released the revision to be decided based upon reviewer 2 and my review of the requested amendments.

I now judge the revised manuscript as sufficiently novel and robust for publication

Reviewer's Responses to Questions

**Key Review Criteria Required for Acceptance?**

**Methods**

-Are the objectives of the study clearly articulated with a clear testable hypothesis stated?

-Is the study design appropriate to address the stated objectives?

-Is the population clearly described and appropriate for the hypothesis being tested?

-Is the sample size sufficient to ensure adequate power to address the hypothesis being tested?

-Were correct statistical analysis used to support conclusions?

-Are there concerns about ethical or regulatory requirements being met?

Reviewer #2: (No Response)

**Results**

-Does the analysis presented match the analysis plan?

-Are the results clearly and completely presented?

-Are the figures (Tables, Images) of sufficient quality for clarity?

Reviewer #2: (No Response)

**Conclusions**

-Are the conclusions supported by the data presented?

-Are the limitations of analysis clearly described?

-Do the authors discuss how these data can be helpful to advance our understanding of the topic under study?

-Is public health relevance addressed?

Reviewer #2: (No Response)

**Editorial and Data Presentation Modifications?**

Reviewer #2: (No Response)

**Summary and General Comments**

Reviewer #2: (No Response)

PLOS authors have the option to publish the peer review history of their article (what does this mean?). If published, this will include your full peer review and any attached files.

Reviewer #2: **Yes: **Moses Bockarie

---

## [Editor Report · Acceptance letter]

17 Dec 2020

Dear Dr. Hodges,

We are delighted to inform you that your manuscript, "Achievements and challenges of lymphatic filariasis elimination in Sierra Leone," has been formally accepted for publication in PLOS Neglected Tropical Diseases.

Best regards,

Shaden Kamhawi

co-Editor-in-Chief

Paul Brindley

co-Editor-in-Chief
